# Effects of Zinc-Biofortified Wheat Intake on Plasma Markers of Fatty Acid Metabolism and Oxidative Stress Among Adolescents [note 1]

**DOI:** 10.3390/nu16244265

**Published:** 2024-12-11

**Authors:** Babar Shahzad, Roberta R. Holt, Swarnim Gupta, Mukhtiar Zaman, Muhammad Shahzad, Nicola M. Lowe, Andrew G. Hall

**Affiliations:** 1Institute of Basic Medical Sciences, Khyber Medical University, Peshawar 25100, Pakistan; babar.kmu@gmail.com (B.S.); shahzad.ibms@kmu.edu.pk (M.S.); 2Department of Nutrition, University of California, Davis, CA 95616, USA; rrholt@ucdavis.edu; 3Centre for Global Development, University of Central Lancashire, Preston PR1 2HE, UK; sgupta6@uclan.ac.uk (S.G.); nmlowe@uclan.ac.uk (N.M.L.); 4Department of Pulmonology, Rehman Medical Institute, Peshawar 25000, Pakistan; mukhtiar.zaman@rmi.edu.pk

**Keywords:** zinc biomarkers, plasma zinc concentration, fatty acid desaturation, oxylipins, cardiometabolic health, biofortification, Pakistan

## Abstract

Background/Objective: Zinc deficiency is common worldwide and has been linked to reduced growth and immune function, increased risk of and slower recovery from infections, and increased risk of non-communicable diseases. To address the issue, zinc biofortification of wheat has been proposed as a sustainable approach to increase dietary zinc intake in countries like Pakistan, where zinc deficiency rates are high and wheat is the primary staple crop. Since plasma zinc concentration (PZC) does not reliably respond to small changes in zinc intake, biomarkers sensitive to small changes in zinc intake achievable though biofortification are needed. Activity indices for zinc-dependent metabolic steps of desaturation and elongation of omega-6 fatty acids (FAs) have been proposed as sensitive zinc biomarkers. Oxylipin metabolites of polyunsaturated FAs may also respond to changes in zinc intake and further mediate metabolic response to oxidative stress. The objective of the current study was to assess the effects of consuming zinc-biofortified wheat flour on plasma markers of fatty acid (FA) metabolism in females aged 10–16 years. Methods: A nested secondary analysis was conducted in samples from a double-blind, cluster-randomized controlled trial conducted in rural Pakistan, whereparticipants (n = 517) consumed either zinc-biofortified wheat flour or control flour for 25 weeks. Total plasma FAs and oxylipins were measured by liquid chromatography tandem mass spectrometry (LC–MS/MS). Activity indices were estimated from the ratios of product to precursor FAs. Results: Except for docosahexaenoic acid (DHA, *p* < 0.05), no significant intervention effect was observed on plasma FAs and FA activity index endpoints. Zinc-biofortified wheat intake reduced pro-inflammatory oxylipins and biomarkers of oxidative stress, 5-HETE (*p* < 0.05), 9-HETE (*p* < 0.05), 11-HETE (*p* < 0.05), and 15-HETE (*p* < 0.05), compared with the control. However, after adjustment for multiple comparisons, none of the intervention effects remained significant. Conclusions: Further study of the responsiveness and specificity of plasma oxylipins to changes in zinc intake is warranted.

## 1. Introduction

Zinc is an essential micronutrient and participates in numerous biochemical pathways crucial for cell growth and development. It is estimated that 17.3% of the global population is at risk of inadequate zinc intake [1], which has consequences in terms of numerous biochemical pathways. The human body contains 2–3 g of zinc, with more than half found in skeletal muscle and organs [2]. Through its roles in protein structure, enzyme active sites, signaling, and regulation, zinc determines a wider range of critical functions than any other single micronutrient. Zinc is well known for its roles in tissue growth and immune function, and its deficiency has been associated with reduced child growth and increased risk of infections [3]. Zinc is also needed for the regulation of metabolism and vascular function. Inadequate zinc intake has been implicated in the risks of type 2 diabetes mellitus and cardiovascular disease [4,5,6].

Zinc deficiency is of global concern. Based on modeling the availability of zinc in national food supplies, the highest risk for zinc deficiency is predicted in low- and middle-income countries (LMICs) of Sub-Saharan Africa and South Asia, often exceeding 25% of the population [1]. In Pakistan, 20% were estimated to have inadequate zinc intakes [1]. However, zinc deficiency rates determined by plasma zinc concentration (PZC) are often higher than rates based on modeling dietary data [7], and are often higher in rural areas and at-risk populations [8,9]. In 2018, 22% of Pakistani women of reproductive age (WRA) were zinc-deficient based on the PZC [10]. In the catchment areas of rural Khyber Pakhtunkhwa, Pakistan, where the present study was conducted, two in every three adolescent females aged 10 to 16 years had a PZC indicative of zinc deficiency [11], emphasizing the need for intervention.

Increasing zinc intake and combating global zinc deficiency, especially among the most vulnerable population groups such as adolescents, involves context-specific strategies that combine complementary evidence-based interventions including supplementation, food fortification, and biofortification [12,13]. Zinc biofortification has been demonstrated to increase zinc intakes and has been proposed as a cost-effective and sustainable approach to alleviate zinc deficiency and improve health [14]. However, due to the relatively small increases in zinc intake achievable with biofortification, a change in PZC may not be readily detected. In our recent study, Biofortified Zinc Flour to Eliminate Deficiency (BiZiFED2), a 1.5 mg/d increase in zinc intake for 25 weeks did not result in a significant effect on PZC in adolescent females [11]. The lack of sensitivity of PZC to detect functionally significant changes in dietary zinc intake has been described previously [15,16]. To summarize, PZC is susceptible to factors unrelated to zinc nutriture, including diurnal variations, stress (both psychological and physical), exercise, and infections. This means that PZC data must be interpreted in the context of these confounding factors or efforts must be made to eliminate them as far as possible. Much research over the last three decades has focused on evaluating alternative, more sensitive and specific biomarkers of zinc status [17].

Novel functional zinc biomarkers have therefore been proposed for the evaluation of zinc status [17,18]. Biomarkers of zinc-dependent metabolic functions, such as omega-6 fatty acid (FA) desaturation, correlate with PZC and may respond to changes in zinc intake more readily than PZC. However, heterogeneity exists between the studies and a recent meta-analysis found no consistent relationship between indices of omega-6 FA desaturation and dietary zinc intakes [19]. FA desaturation is further linked to the regulation of inflammation through oxylipin metabolites of linoleic and arachidonic acid [20]. Moderate changes in zinc intake have been demonstrated to alter the balance between pro-inflammatory and anti-inflammatory oxylipins [21].

The objective of the present study was to conduct a nested secondary analysis of the effect of consuming zinc-biofortified wheat flour on biomarkers of lipid metabolism, specifically FAs and oxylipins in female adolescents participating in the BiZiFED2 cluster-randomized controlled trial in rural Khyber Pakhtunkhwa, Pakistan. We hypothesized that 25 weeks of increased zinc intake from zinc-biofortified wheat would increase the desaturation of omega-6 FAs and shift oxylipins to a pattern consistent with reduced inflammation, compared with control wheat. These data have potential impact towards the identification and future study of biomarkers of zinc function that may respond to changes in zinc intake in the context of zinc biofortification.

## 2. Materials and Methods

### 2.1. Study Setting

This study was nested within the BiZiFED2 cluster-randomized, double-blind controlled trial. The parent study was designed to examine the effectiveness of consuming zinc-biofortified wheat flour on zinc status and health outcomes in adolescent females living in rural Khyber Pakhtunkhwa, Pakistan [22]. As neighboring households in these communities often share food, a cluster-randomized design was chosen to minimize contamination between intervention and control groups while ensuring robust evaluation under real-world conditions. Wheat was selected as the vehicle for introducing additional zinc into the diet through biofortification due to its ubiquity in the local diet. Wheat is the primary staple crop in this region and is typically consumed with every meal in the form of bread (roti, chapati, naan, and paratha). In this community, bread is baked daily within each household using locally procured wheat flour; thus, flour was selected as the vehicle for introducing additional zinc into the diet through biofortification. In 2016, a new variety of wheat, Zincol-2016, was released in Pakistan that was selectively bred for its high grain zinc content. The parent study was the first effectiveness study of the impact of consuming flour milled from Zincol-2016 in a free-living community setting. The primary hypothesis of the BiZiFED2 RCT was that consuming zinc-biofortified wheat flour would lead to an improvement in plasma zinc concentration (PZC), thereby reducing zinc deficiency among adolescent females. The secondary hypotheses included improvements in hematological indices, such as hemoglobin and iron status, as well as other functional outcomes and potential biomarkers of zinc. This paper specifically focuses on secondary analyses related to lipid metabolism and markers of inflammation.

### 2.2. Ethical Approval

Ethical approval was granted from the University of Central Lancashire STEMH Ethics Committee (reference number: STEMH 1014) and the Khyber Medical University Ethics Committee (reference number: DIR/KMU-EB/BZ/000683). This study was registered with the ISRCTN registry (Trial registration number ISRCTN17107812). The design and conduct of the BiZiFED2 RCT adhered to the Consolidated Standards of Reporting Trials (CONSORT) guidelines for cluster-randomized analyses as detailed in the publication on the primary outcome [11].

### 2.3. Participants, Study Design, and Sample Selection

The recruitment and cluster randomization strategies and trial protocol have been previously described [11,22]. Briefly, the households from two adjacent communities comprising a total of 44 clusters (hamlets) were assessed for eligibility using criteria households with at least one unmarried, non-pregnant, and non-lactating adolescent girl (10–16 years) and one child (1–5 years). No other criteria were included in this effectiveness trial. Household eligibility was recorded and used in the cluster selection process. Clusters were arranged sequentially based on mean household size, starting with the smallest, until the target sample size of 500 adolescent–child pairs was achieved. This sample size was powered for the primary outcome of the trial and has been detailed in the study protocol [22].

For obtaining the consent, the head of each household was first approached, as is culturally appropriate, and the study purpose was explained. If the household head consented, the adolescent girl and the mother of the child were individually approached, and participant information sheets were provided in the local language (Pushto) along with verbal explanations to ensure comprehension. Consent was recorded either through initials or by marking an ‘X’ for those unable to write. A total of 517 adolescent–child pairs from 486 households across 34 clusters were ultimately enrolled between September 2019 and November 2019. These clusters were then paired based on household size and the age of the adolescent girl to ensure comparability at baseline followed by random allocation to intervention or control arms performed using computer-generated software, with each cluster having an equal probability of receiving either biofortified or control flour. The randomization allocation was known only to the study director, ensuring that all field team members, participants, and data analysts remained blinded to the intervention allocation. Details on the implementation of masking, including procedures during storage, milling, and throughout distribution, have been described in detail elsewhere [11].

Characteristics of the participants, including age and indicators of socioeconomic status, were collected at enrollment. In Phase I (equilibration period), all households were provided with the control (Galaxy) wheat flour. Blood samples and data such as 24-h dietary recalls were collected at the end of this phase (late August–early September 2020) to establish a robust baseline before the intervention. During Phase II (intervention phase), which commenced on 22 September 2020, participating households received either the control or biofortified (Zincol-2016) flour henceforth referred to as control or zinc-biofortified wheat (ZBW) flour, respectively, for 25 weeks.

Participant information was de-identified using unique codes, with identifiers securely stored separately from study data. Biological samples were securely labelled and shipped using a traceable courier service, while data transfer was conducted via password-protected files. Further, participant safety was ensured through oversight by the Study Advisory Group (SAG) and an independent Data Monitoring Committee (DMC). The SAG provided strategic guidance on ethical and operational matters, while the DMC monitored study progress and reviewed adverse events, thus maintaining participant safety and study integrity. These measures are detailed elsewhere [22].

To ensure a robust and unbiased study design, the following steps were undertaken in the sample selection protocol for the present study. A total of 399 participants who completed the endline were considered for the analysis of plasma markers of fatty acid metabolism. Their baseline and 25-week plasma samples were shipped to the University of California (UC), Davis. For oxylipin measures, due to limited funds, a subsample of 100 participants (50 per intervention arm) were randomly selected, with equal probability of being selected from each intervention group using Random Allocation Software (https://mahmoodsaghaei.tripod.com/Softwares/randalloc.html, accessed on 1 June 2021) [23], from the above 399 participants samples.

### 2.4. Sample Collection, Storage, and Transportation

There were no pre-requirements such as fasting, specific timing, physical activity, or medication restrictions for blood collection. Whole blood (non-fasted) was drawn from the antecubital vein through a butterfly needle into trace-element-free vacuum collection tubes containing EDTA anticoagulant (BD, Franklin Lakes, NJ, USA). The plasma was separated within 40 min of collection by centrifugation for 4 min at 2500× *g*. Samples were transported on ice to Khyber Medical University and stored at −80 °C. For transportation to UC Davis, samples were packed with dry ice in insulated shipping containers designed for long-distance transport of biological specimens. To monitor temperature stability and ensure temperature was maintained during transit, real-time temperature loggers were included in the shipment. Upon arrival at UC Davis, temperature logs confirmed that samples remained below −60 °C throughout transit.

### 2.5. Laboratory Methods

Plasma FAs and cholesterol were analyzed by OmegaQuant (Sioux Falls, SD, USA) as previously described [24]. Briefly, lipids were isolated from plasma samples and subjected to alkaline hydrolysis; then, total FAs were converted to methyl esters (FAMEs) for analysis. FAs were quantified by liquid chromatography tandem mass spectrometry (LC–MS/MS) analysis of a panel of 24 FAs, using a multi-point calibration curve for each FA. The panel included myristic, palmitic, palmitelaidic, palmitoleic, stearic, elaidic, oleic, linoelaidic, linoleic, arachidic, γ-linolenic, eicosenoic, α-linolenic, eicosadienoic, behenic, dihomo-γ-linolenic, arachidonic, lignoceric, eicosapentaenoic, nervonic, adrenic, docosapentaenoic-n6, docosapentaenoic-n3, and docosahexaenoic acids. Activity indices were determined as the ratio of product to precursor FAs covering single or multiple enzymatic steps of desaturation or elongation. The cholesterol panel, determined by autoanalyzer (HORIBA ABX, Montpellier, France), included high-density lipoprotein (HDL) cholesterol, low-density lipoprotein (LDL) cholesterol, total cholesterol, and total triglycerides. Oxylipins were determined at the UC Davis West Coast Metabolomics Center as previously described [25,26]. Briefly, following extraction of FAMEs, esterified oxylipins were transformed by base hydrolysis into oxylipin free acids, isolated by solid-phase extraction, reconstituted in methanol with an internal standard, and quantified with multi-point calibration curves by LC–MS/MS using a Thermo Fisher Vanquish (Thermo Fisher Scientific, Waltham, MA, USA) and SciEx Qtrap 6500+ (SciEx, Framingham, MA, USA). A panel of 68 oxylipins was measured. Stability and assay variability were monitored by analyzing laboratory reference materials in replicate in each batch.

### 2.6. Statistical Analysis

This nested study was analyzed on a per-protocol basis, including only participants who completed the study. Consequently, missing data were not addressed, as the analysis was restricted to completers. Prior to the analysis, potential outliers were assessed using Huber robust fit analysis, with one individual sample removed as this initial screening demonstrated the presence of at least one outlier for several variables. The remaining data were log transformed prior to mixed model analysis, using time and group as fixed factors, participant as a random effect and adjusted for age, BMI, CRP, AGP, triglycerides, and menarche status. Significant interactive effects (i.e., group × time) were corrected for multiple comparisons using Tukey post hoc testing with significance determined at *p* < 0.05. A significant group × time interaction indicated the change over time for that variable differed significantly between groups, thus indicating an effect of the intervention. We applied the Benjamini–Hochberg (BH) method to adjust for the false discovery rate (FDR) for all biomarkers, with significance set at q < 0.05. FDR values that no longer remained significant after correction were reported. Data are presented as the geometric mean [range].

## 3. Results

### 3.1. General Characteristics of the Population

We have previously published a detailed CONSORT flow diagram for the parent RCT along with primary outcomes, demographic data, socioeconomic data, and general characteristics of the participants [11]. Briefly, the study participants were adolescent females and the average age at the point of enrollment (n = 517) was 12.1 ± 1.7 years with a range of 8.6–15.3 years. The majority (61.2%) of households had income below the poverty level of 20,000 PKR. A total of 68.8% (289 of the 420 participants) were zinc-deficient as indicated by a PZC below the cutoff for zinc deficiency. Participant dropout across study points was independent of study arm allocation.

### 3.2. Plasma Lipids and Oxylipins

At baseline, the two treatment groups differed in 6 of the 53 lipid parameters (Table 1, Table 2 and Table 3). Over the course of the intervention, LDL, HDL, and total cholesterol increased by 12.5%, 8.5%, and 9.4%, respectively (Table 1). Similarly, the concentrations of 6 of the 24 FAs increased from baseline after the 25-week intervention: C18:2n6 (LA, 11%), C18:3n3 (ALA, 9%), C20:4n6 (ARA, 10%), C22:5n3 (docosapentaenoic-n3, 7%), C22:6n3 (DHA, 28%), and C24:0 (lignoceric, 5%). In contrast, concentrations of 7 of the 24 FAs decreased: C14:0 (myristic, 14%), C16:0 (palmitic, 4%), C16:1n7 (palmitoleic, 15%), C18:1n9 (oleic, 6%), C18:3n6 (GLA, 10%), C22:4n6 (adrenic, 4%), and C24:1n9 (nervonic, 8%). A significant group and time interaction was observed for DHA with the intake of ZBW for 25 weeks compared to control wheat intake. Although DHA increased in both groups over the intervention period, the extent of the increase in the ZBW group was greater than the control by more than 20% (Table 1). The intervention effect was no longer significant after FDR correction (q = 0.41). No other significant interaction effects were observed for the remaining fatty acids.

There were no significant effects of ZBW intake on any of the FA activity indices covering the various steps of FA desaturation and elongation, compared with the control. However, a number of calculated activity indices showed marked changes over the course of the 25-week intervention (Table 2). From baseline, the estimated activity index for FA elongase (ELOVL) 5,2 (estimated as the ratio C22:5n3/20:5n3) did not significantly change over the 25-week intervention period; however, the omega-6 ELOVL 5,2 activity index was significantly reduced from baseline. In addition, significant decreases in both steroyl-CoA desaturase 1 (SCD1) indices and Δ6 desaturase (C18:3n6/C18:2n6) were observed, while ELOVL 5 (C20:3n6/C18:3n6) and 1,6 (C18:0/C16:0), along with the omega-3 and 6 products of elongation/Δ6 desaturase/beta-oxidation (C22:6n3/C22:5n3 and C22:5n6/C22:4n6, respectively), were significantly increased from baseline.

Of the 68 oxylipins that were measured, 15 had concentrations above the level of detection for evaluation of intervention effects and are therefore presented in Table 3. Over the 25 weeks of dietary intervention, decreases in 9 of these 15 oxylipins were observed. Six were products of ARA via lipoxygenase (5-HETE, 8S-HETE, 12S-HETE, and 15-HETE), cyclooxygenase (11-HETE), or non-enzymatic oxidation (9-HETE). Three were lipoxygenase products of LA (9-HODE, 9-KODE, and 13-HODE). Significant group × time interaction in response to ZBW was observed for the four pro-inflammatory oxylipin products of ARA: 5-HETE, 15-HETE, 11-HETE, and 9-HETE. However, these pro-inflammatory oxylipins were significantly reduced in both groups from baseline over the course of the intervention (Table 3).

## 4. Discussion

Biofortified Zinc Flour to Eliminate Deficiency (BiZiFED) is a research program, launched in 2017, to demonstrate the impact of consuming zinc-biofortified wheat flour on biochemical and functional biomarkers of zinc status in a zinc-deficient population [11]. Biofortification of a staple food is potentially a low-cost, sustainable approach of increasing zinc intake. In order to evaluate the impact of such diet-based interventions to improve zinc intakes, it is important to have a biomarker of zinc status that is sensitive to the modest changes in zinc intake achievable through biofortification. This study explored the potential of plasma FAs and oxylipin metabolites to indicate a functional response to the incremental 1.5 mg/d change in dietary zinc intake among female adolescents who completed the BiZiFED2 cluster-randomized controlled trial in rural Khyber Pakhtunkhwa, Pakistan. This is the first study we are aware of reporting plasma FA response to a dietary zinc intervention in an adolescent population and only the second human study to measure oxylipin response to changes in zinc intake [21].

The relevance of zinc nutritional status to polyunsaturated FA metabolism, specifically the conversion of LA to ARA, and further conversion to oxylipins, was initially proposed more than four decades ago [28]. More recently, the LA:DGLA ratio was put forth as a putative biomarker of zinc nutritional status due to its correlation with PZC [29,30]. Although randomized studies of zinc supplementation or fortification in humans have had variable results in the magnitude of plasma FA response, the activity indices describing Δ6-desaturase activity can have a greater response to changes in dietary zinc intake compared with the PZC [19].

### 4.1. Key Findings

The objective of the present study was to conduct a nested secondary analysis of the effect of consuming zinc-biofortified wheat flour on FAs and oxylipins. In response to 25 weeks consuming the flour, plasma concentrations of the omega-3 FA, DHA, increased and plasma concentrations of four pro-inflammatory oxylipin metabolites of ARA decreased due to the intake of ZBW. However, after adjusting for multiple comparisons, no significant intervention effects remained. Furthermore, no statistically significant effects of the intervention were observed on other plasma lipid biomarkers, including activity indices covering the desaturation and elongation of LA to ARA.

Over the course of the 25-week intervention period, the study population showed marked changes in numerous plasma lipid endpoints that were independent of group assignment. While LDL, HDL, and total cholesterol all increased, about half of the FAs that were measured changed over the course of the intervention, with six increasing and seven decreasing. This included a reduction in FA activity indices of stearoyl-CoA desaturase and palmitic acid, with a corresponding increase in circulating levels of α-linolenic acid and linoleic acid and their respective longer chain products, docosahexaenoic and docosapentaenoic acids, respectively. In contrast, all of the oxylipins that changed over the 25-week intervention period decreased. A significant reduction in lipoxygenase, cyclooxygenase, and non-enzymatic oxylipins, but not cytochrome P450 derived oxylipins, was observed.

Since plasma lipids are largely reflective of their dietary intake, their changes over the course of 25 weeks may have been influenced by seasonal changes in the diets. In a study of the influence of season on dietary intakes among urban adolescents in Pakistan, energy from fat had the highest seasonal variation of energy sources and was the highest in the autumn and winter compared with summer and spring [31]. In the BiZiFED2 samples analyzed here, baseline was collected at the end of summer 2020, and the 25-week intervention period ended near the end of winter 2021. Although data on seasonal changes in circulating FAs and oxylipins are scarce, one study reported a strong seasonal influence on circulating oxylipins in a small sample (n = 9) of patients with seasonal depression [32].

### 4.2. Limitations

Since the analysis was per-protocol, these observations are only generalizable to the population that completed the parent study. The ZBW of the parent study effected a 1.5 mg/d increase in dietary zinc intake compared with the control, although the initial expectation based on pilot studies was an increase of 3.3 mg/d [11]. The modest increase in zinc intake is not necessarily a limitation in the context of this study, since the overarching aim of the BiZiFED trial was to explore what was feasible under real-world conditions, i.e., an effectiveness trial. The level of biofortification obtained through the wheat breeding and agronomy methods was low (newer biofortified varieties have improved upon this); however, measurable changes in the oxylipin profiles are still notable. Other biomarkers of zinc status were also explored in the broader BiZiFED trial and have been reported elsewhere [33].

The ability of the current study to detect a functional change relevant to cardiometabolic health warrants further investigation using a study designed and powered to specifically explore this as a primary outcome for a range of zinc intake levels. The limitation in the collection of other biomarkers related to cardiovascular health, such as blood pressure, further constrained the present study.

### 4.3. Mechanisms and Comparison with Other Studies

DHA is of critical importance to neuronal development [34]. DHA and the activity of the desaturation pathway of ALA to DHA have further been implicated in non-communicable diseases including type 2 diabetes mellitus and cardiovascular disease, although less than 1% of DHA is estimated to be derived from the conversion of ALA in humans [35]. Suh et al. (2022) also reported no significant effect of increased zinc intake on DHA; however, increases in the omega-3 FAs eicosatetraenoic acid C20:4n–3, eicosapentaenoic acid C20:5n–3, and docosapentaenoic acid C22:5n–3 were observed [21]. In a study of patients with type 2 diabetes, no changes in erythrocyte DHA content were observed in response to zinc supplementation [36]. Moreover, in healthy Beninese children aged 6 to 10 years, there was no significant effect of consuming zinc-fortified water on total plasma DHA [37].

Oxylipins are important mediators of cellular inflammatory (and anti-inflammatory) responses that have been linked with the risk and progression of non-communicable diseases [38,39,40]. In limited data from pediatric populations where oxylipins were measured, pro-inflammatory oxylipins had generally higher plasma concentrations with age and were associated with dietary intake of precursor FAs and environmental exposure to tobacco smoke [41]. Additionally, pro-inflammatory oxylipins in umbilical cord blood were predictive of adverse infant health outcomes, including bronchopulmonary dysplasia and neonatal hypertension [42].

Data from recent studies of oxylipins and zinc nutriture are further suggestive of their mechanistic connection at the cellular level. A murine knockout model of the zinc transporter, Znt7, led to elevated oxylipin products of LA and ARA, indicating a connection between pro-inflammatory oxylipins and transporter-mediated zinc metabolism [43]. In humans, the recently-identified Ziegler–Huang syndrome, is caused by loss of function of the analogous hZnT7 transporter [44]. The testicular hypoplasia observed in Ziegler–Huang syndrome is thought to be related to the particular sensitivity of the organ to oxidative stress induced by free radicals [44]. A primary role of hZnT7 is the transport of zinc ions into the Golgi apparatus, where key function-determining modifications to nascent proteins are accomplished [45]. It is therefore plausible that loss of hZnT7 function excludes zinc from protein-functional roles, disproportionate to whole body zinc nutritional status. The resulting stunted growth, bone marrow failure, and testicular hypoplasia in two adolescent cases, was accompanied by only moderately depressed PZC [44].

Oxylipins were related with zinc nutriture in one previous human study. In response to dietary zinc repletion at 10 mg zinc per day following a period of low zinc intake, significant increases in anti-inflammatory oxylipin metabolites of EPA (14(15)-EpETE and 11(12)-EpETE), and decreases in the pro-inflammatory oxylipin metabolite of LA (12,13-Ep-9-KODE), were observed [21]. When the daily zinc intake was increased by 25 mg for 3 weeks using a zinc supplement, increases in the anti-inflammatory metabolites 14(15)-EpETE and 11(12)-EpETE became more apparent.

In the present study, pro-inflammatory oxylipin products of lipoxygenase (5-HETE and 15-HETE), cyclooxygenase (11-HETE), and non-enzymatic oxidation (9-HETE) of ARA were reduced to a greater degree among participants consuming ZBW. Although the specific mechanisms remain to be elucidated, these results demonstrate the potential sensitivity of oxylipins to modest changes in zinc intake in adolescents, which is of particular interest towards the identification of biomarkers that may be used for future evaluation of zinc biofortification efforts.

The relation of oxylipins to zinc metabolism and oxidative stress is also intriguing. Total HETEs has been used as a biomarker of oxidative stress in widely divergent clinical conditions from the progression of Parkinsonism [46] to determining the extent of oxidative damage following Dengue Fever infection [47]. Each of the four HETEs that were affected by ZBW intake have individually been used as biomarkers of oxidative stress in human studies: 9-HETE indicated oxidative stress in coronary artery disease patients [48]. 9-, 11-, and 15-HETE were elevated in children and young adults with oxidative stress due to polycystic kidney disease [49]. 5-HETE correlated with cardiometabolic risk in young adults [50]. More recently, 12- and 15-HETE were implicated in the risk of preterm birth with preeclampsia, highlighting their relevance to reproductive health [51]. These data are therefore suggestive of reduced oxidative stress due to ZBW intake in the BiZiFED2 population, potentially relevant to both cardiometabolic and reproductive health.

In Suh et al., increases in Δ6-desaturase and decreases in Δ5-desaturase activities were also observed in response to the intervention, although the changes were not always congruent with changes in the oxylipins. Interestingly, the changes in desaturase activity indices in Suh et al. reverted to baseline when zinc intakes were increased by 25 mg [21]. These data, along with data from the present study, further demonstrate the heterogeneity of polyunsaturated FA response to zinc. Although sensitive to depletion and repletion of dietary zinc, FAs may not respond in the same way between low and relatively high zinc intakes. It is also plausible that the method of provision (as a supplemental pill, vs. as a component of food) may have influenced the effects on FA desaturation [52]. This is a potentially important consideration when selecting biomarkers of effect for food-based approaches to increase zinc intake.

## 5. Conclusions

This study was conducted in a low resource community in rural Pakistan where the zinc deficiency rate in adolescent females is more than twice as high as previously observed in women of reproductive age (WRA) [53]. Although few studies are available, existing data suggest that numerous micronutrients, including zinc, may be deficient in the diets of Pakistani adolescents [11,53,54,55]. Adequate zinc intake is critical in all stages of life but especially during adolescence, towards normal growth and development, and also to lay the foundations of future health.

Thus, the implications of the findings of this study are two-fold. Firstly, this study has demonstrated that a modest increase in dietary zinc intake is reflected in detectable changes in oxylipin pathways. This warrants further investigation as a potential biomarker for monitoring the impact of food-based interventions on zinc status. Secondly, inadequate dietary zinc intake increases inflammation, oxidative stress, and the risk of cardiovascular disease. Rapid increases in cardiovascular disease in Pakistan are expected to continue [56]. Zinc deficiency has also been associated with high blood pressure in US adolescents [57]. Non-communicable disease risk is a growing concern for adolescents globally, particularly in LMICs [58].

Dysregulated oxylipin pathways are predictive of cardiometabolic risk and may contribute to adverse reproductive health outcomes. Understanding the role of oxylipins in immune and inflammatory responses further underscores their significance in the prevention and management of CVD [59], while non-communicable diseases are increasingly linked with zinc nutriture [6,60,61]. The BiZiFED2 ZBW intervention had a tendency to reduce oxylipin biomarkers of oxidative stress that have further been implicated in adverse cardiometabolic and reproductive health outcomes. Importantly, these outcomes demonstrate potential indicators of cardiometabolic health that may be responsive to the modest increases in zinc intakes that may be achievable through biofortification alone. Future research into strategies for increasing zinc nutriture, particularly in populations with elevated cardiometabolic or reproductive health risks, should therefore consider the inclusion of oxylipin measures of effect.

## Figures and Tables

**Table 1 nutrients-16-04265-t001:** Plasma cholesterol, triglycerides, and fatty acids.

	Control (Group 1)		ZBW (Group 2)		*p*-Values		
	Baseline (n = 212)	25 Weeks (n = 212)	Baseline (n = 187)	25 Weeks (n = 186) ^1^	Time	Group	Interaction
Total cholesterol	123.1 [222.0]	136.3 [188.0]	118.4 [226.0]	128.0 [180.0]	<0.001	0.01	0.36
LDL cholesterol ^2^	67.4 [131.2]	76.9 [156.9]	61.2 [153.9]	67.8 [132.9]	<0.001	<0.001	0.34
HDL cholesterol	41.7 [67.7]	45.4 [52.3]	41.7 [64.3]	45.1 [45.9]	<0.001	0.83	0.95
Triglycerides	81.8 [255]	81.4 [179.0]	78.8 [229.0]	81.7 [160.0]	0.70	0.97	0.18
C14:0 (myristic)	21.0 [112.1]	18.2 [86.3]	19.4 [105.2]	16.4 [113.9]	<0.001	0.06	0.27
C16:0 (palmitic)	649.2 [2132.3]	617.7 [1346.9]	628.3 [2193.8]	606.5 [2048.9]	<0.01	0.52	0.76
C16:1n7 (palmitoleic)	59.6 [217.0]	50.3 [212.1]	53.1 [289.4]	45.5 [314.3]	<0.001	0.02	0.64
C16:1n7t (palmitelaidic) ^2^	3.86 [22.7]	3.60 [15.5]	3.16 [24.5]	3.16 [14.4]	0.44	<0.01	0.48
C18:0 (stearic)	164.8 [393.7]	164.6 [293.8]	159.9 [420.2]	162.7 [351.8]	0.91	0.53	0.83
C18:1n9t (elaidic)	17.0 [259.8]	14.2 [120.6]	12.6 [348.9]	13.1 [137.3]	0.53	0.02	0.06
C18:1n9 (oleic)	643.3 [1968.3]	594.0 [1424.8]	616.7 [2084.0]	587.7 [2135.3]	<0.001	0.70	0.77
C20:0 (arachidic)	2.26 [12.4]	2.22 [8.71]	2.36 [6.33]	2.27 [7.66]	0.32	0.42	0.41
C20:1n9 (eicosenoic)	5.13 [14.9]	4.85 [11.5]	4.93 [15.0]	5.09 [13.7]	0.62	0.83	0.06
C22:0 (behenic)	4.22 [8.90]	4.35 [7.94]	4.42 [9.02]	4.48 [9.19]	0.09	0.30	0.69
C24:0 (lignoceric)	3.00 [7.68]	3.16 [8.93]	3.21 [9.36]	3.36 [9.21]	0.01	0.14	0.80
C24:1n9 (nervonic)	4.64 [21.3]	4.12 [14.4]	4.68 [16.5]	4.43 [12.7]	<0.01	0.48	0.13
C18:2n6 (LA)	677.0 [1478.3]	743.0 [1596.9]	656.8 [1472.8]	734.1 [1418.3]	<0.001	0.45	0.59
C18:2n6t (linoelaidic)	11.4 [54.4]	11.0 [58.7]	10.3 [44.6]	10.9 [40.6]	0.68	0.21	0.19
C18:3n6 (GLA)	13.4 [49.7]	11.8 [37.1]	12.8 [46.9]	11.5 [50.7]	<0.001	0.37	0.91
C20:2n6 (eicosadienoic)	6.25 [14.0]	6.08 [16.7]	5.74 [13.0]	6.04 [12.4]	0.63	0.17	0.08
C20:3n6 (DGLA) ^1^	47.6 [117.9]	46.0 [116.0]	43.5 [116.3]	43.7 [113.8]	0.054	0.02	0.36
C20:4n6 (ARA)	130.5 [260.5]	142.9 [371.3]	122.6 [262.5]	136.4 [284.4]	<0.001	0.06	0.69
C22:4n6 (adrenic)	7.24 [18.3]	6.86 [18.3]	6.93 [20.3]	6.71 [22.0]	<0.001	0.29	0.83
C22:5n6 (docosapentaenoic-n6)	7.81 [22.8]	8.03 [28.6]	7.57 [17.3]	7.69 [18.9]	0.65	0.44	0.38
C18:3n3 (ALA)	9.34 [69.1]	9.72 [59.1]	9.01 [59.9]	10.3 [72.2]	0.01	0.83	0.26
C20:5n3 (EPA)	5.83 [19.4]	5.97 [30.5]	5.34 [21.6]	5.68 [19.6]	0.52	0.07	0.57
C22:5n3 (docosapentaenoic-n3)	9.15 [25.7]	9.59 [17.7]	8.43 [19.2]	9.17 [21.2]	0.02	0.06	0.52
C22:6n3 (DHA) ^1^	22.6 [46.8]	28.1 [59.7]	20.4 [55.7]	27.1 [45.0]	<0.001	0.02	0.03 ^3^

Legend: Data are expressed as the geometric mean [range]. Units are µg/mL for all FAs and µg/dL for cholesterol and triglycerides. Abbreviations are ALA, α-linolenic acid; ARA, arachidonic acid; DGLA, dihomo-γ-linolenic acid; DHA, docosahexaenoic acid; EPA, eicosapentaenoic acid; GLA, γ-linolenic acid; HDL, high-density lipoprotein; LA, linoleic acid; LDL, low-density lipoprotein; ZBW, zinc-biofortified wheat. ^1^ One sample was removed following detection of outliers in multiple variables. ^2^ Significantly different at baseline between groups. ^3^ Not significant after false discovery rate (FDR) correction.

**Table 2 nutrients-16-04265-t002:** Fatty acid desaturase and elongase activity indices.

	Control (Group 1)		ZBW (Group 2)		*p*-Values		
	Baseline (n = 212)	25 Weeks (n = 212)	Baseline (n = 187)	25 Weeks (n = 186) ^1^	Time	Group	Interaction
C16:1n7/C16:0 (SCD1) ^2^	0.092 [0.222]	0.081 [0.155]	0.085 [0.171]	0.075 [0.148]	<0.001	0.01	0.73
SCD1 (C18:1n9/C18:0)	3.85 [3.21]	3.61 [4.45]	3.86 [3.98]	3.61 [4.25]	<0.001	0.78	0.41
Δ6 desaturase (C18:3n6/C18:2n6)	0.020 [0.073]	0.016 [0.047]	0.020 [0.076]	0.016 [0.052]	<0.001	0.69	0.84
Δ6 desaturase+ELOVL5 (C20:3n6/C18:2n6) ^2^	0.070 [0.122]	0.062 [0.105]	0.066 [0.087]	0.060 [0.113]	<0.001	0.07	0.55
ELOVL5 (C20:3n6/C18:3n6)	3.55 [9.60]	3.89 [9.05]	3.37 [7.10]	3.79 [10.6]	<0.001	0.20	0.34
Δ5 desaturase (C20:4n6/C20:3n6)	2.74 [4.74]	3.11 [5.72]	2.82 [5.62]	3.12 [6.21]	<0.001	0.57	0.45
ELOVL1,6 (C18:0/C16:0)	0.25 [0.24]	0.27 [0.24]	0.25 [0.29]	0.27 [0.30]	<0.001	0.88	0.29
ELOVL5,2 (C22:5n3/C20:5n3)	1.57 [4.25]	1.61 [3.51]	1.58 [5.05]	1.61 [2.17]	0.12	0.55	0.92
ELOVL5,2 (C22:4n6/C20:4n6)	0.056 [0.073]	0.048 [0.091]	0.057 [0.074]	0.049 [0.079]	<0.001	0.39	0.89
C22:5n6/C22:4n6 ^3^	1.08 [1.72]	1.17 [1.81]	1.09 [1.90]	1.15 [2.11]	<0.001	0.90	0.39
C22:6n3/C22:5n3 ^3^	2.45 [5.00]	2.93 [6.89]	2.42 [4.04]	2.95 [5.00]	<0.001	0.72	0.25

Legend: Data are expressed as the geometric mean [range]. Abbreviations are ELOVL, fatty acid elongase; SCD1, steroyl-CoA desaturase 1; ZBW, zinc-biofortified wheat. ^1^ One sample was removed following detection of outliers in multiple variables. ^2^ Significantly different at baseline between groups. ^3^ ELOVL2,4/Δ6 desaturase/beta-oxidation.

**Table 3 nutrients-16-04265-t003:** Plasma oxylipins.

		Control (Group 1)		ZBW (Group 2)		*p*-Values		
Oxylipin	Precursor FA	Baseline (n = 50)	25 Weeks (n = 50)	Baseline (n = 50)	25 Weeks (n = 50)	Time	Group	Interaction
9,10-DiHOME	C18:2n6	0.38 [1.36]	0.43 [3.42]	0.37 [1.41]	0.36 [1.62]	0.54	0.23	0.51
12,13-EpOME	C18:2n6	7.53 [30.9]	7.99 [49.2]	6.74 [23.1]	7.09 [45.4]	0.65	0.55	0.97
12,13-DiHOME	C18:2n6	7.17 [28.6]	7.37 [26.6]	7.18 [22.3]	6.47 [27.6]	0.69	0.56	0.53
9-HODE	C18:2n6	78.7 [253.2]	17.0 [112.0]	89.2 [322.7]	15.8 [46.9]	<0.001	0.4	0.23
9-KODE	C18:2n6	1.12 [2.97]	0.46 [1.52]	1.19 [4.04]	0.46 [1.54]	<0.001	0.5	0.69
13S-HODE	C18:2n6	89.5 [327.6]	19.5 [128.3]	101.5 [347.5]	16.7 [49.4]	<0.001	0.59	0.09
5,6-DiHETrE	C20:4n6	0.71 [1.38]	0.84 [2.10]	0.73 [2.07]	0.83 [2.18]	0.06	0.98	0.55
14,15-DiHETrE	C20:4n6	5.47 [14.8]	5.40 [11.8]	5.24 [10.1]	4.85 [9.91]	0.37	0.13	0.57
5-HETE	C20:4n6	39.4 [195.7]	7.06 [58.9]	44.4 [155.8]	5.58 [27.5]	<0.001	0.98	0.03 ^1^
8S-HETE	C20:4n6	12.3 [87.7]	0.0 [28.8]	12.5 [91.7]	0.0 [4.08]	<0.001	0.5	0.09
11-HETE	C20:4n6	13.6 [98.2]	1.32 [32.5]	15.2 [99.7]	0.96 [5.06]	<0.001	0.96	0.04 ^1^
12S-HETE	C20:4n6	17.9 [134.4]	1.88 [42.0]	19.7 [129.9]	1.54 [5.60]	<0.001	0.8	0.14
15-HETE	C20:4n6	23.8 [161.6]	2.99 [64.0]	27.4 [167.6]	2.16 [8.82]	<0.001	0.94	0.02 ^1^
9-HETE	C20:4n6	16.7 [128.4]	1.61 [41.8]	19.5 [149.6]	1.17 [5.31]	<0.001	1	0.03 ^1^
15,16-DiHODE	C18:3n3	15.7 [51.4]	14.6 [281.0]	14.4 [72.4]	13.9 [136.1]	0.72	0.45	0.77

Legend: Data are expressed as the geometric mean [range]. All concentrations are in nmol/L. Abbreviations are 9,10-DiHOME, 9,10-dihydroxyoctadec-12-enoic acid; 12,13-EpOME, 11-(3-pentyloxiran-2-yl)undec-9-enoic acid; 12,13-DiHOME, 12,13-dihydroxyoctadec-9-enoic acid; 9-HODE, 9-hydroxyoctadeca-10,12-dienoic acid; 9-KODE, 9-oxooctadeca-10,12-dienoic acid; 13S-HODE, 13-hydroxyoctadeca-9,11-dienoic acid; 5,6-DiHETrE, 5,6-dihydroxyicosa-8,11,14-trienoic acid; 14,15-DiHETrE, 14,15-dihydroxyicosa-5,8,11-trienoic acid; 5-HETE, 5-hydroxyicosa-6,8,11,14-tetraenoic acid; 8S-HETE, 8-hydroxyicosa-5,9,11,14-tetraenoic acid; 11-HETE, 11-hydroxyeicosatetraenoic acid; 12S-HETE, 12-hydroxyicosa-5,8,10,14-tetraenoic acid; 15-HETE, 15-hydroxyicosa-5,8,11,13-tetraenoic acid; 15,16-DiHODE, 9-HETE, 9-hydroxyicosa-5,7,11,14-tetraenoic acid; 15,16-dihydroxyoctadeca-9,12-dienoic acid; ZBW, zinc-biofortified wheat. ^1^ Not significant after false discovery rate (FDR) correction. Preliminary results of this table were presented in [27].

## Data Availability

The raw data supporting the conclusions of this article will be made available by the authors on request to preserve the privacy of the participants.

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
