# Peer review of "Effects of Zinc-Biofortified Wheat Intake on Plasma Markers of Fatty Acid Metabolism and Oxidative Stress Among Adolescentsâ€"

_nutrients, 2024, doi:10.3390/nu16244265_

Round 1
Reviewer 1 Report
Comments and Suggestions for Authors
Dear authors, I congratulate you on conducting this study, which is of great relevance to public health. However, I have some comments and issues regarding the material presented.
- I recommend revising the abstract to remove background information and include more details on methods and results. Additionally, the objective of the study is not addressed in the abstract.
- In the methods section, the authors state that they followed the CONSORT guidelines for conducting the study, but I could not find the completed checklist among the manuscript files. I recommend including it as supplementary material to make the process more transparent for the readers.
- The reference indicated for the protocol [6] (line 107) does not correspond to the information described in the text; please revise.
- Regarding the statistical analysis, it is unclear what strategy was used—whether it was intention-to-treat or per-protocol. The manuscript does not provide the study flowchart, including the number of clusters/individuals included, nor how many underwent the final evaluation. Also, the number of dropouts and reasons for them are not reported. Including this information is extremely important, even if it is in the supplementary material. It is also important to describe the strategy adopted for dealing with missing data if the analysis was intention-to-treat. Finally, was there any difference between those who dropped out of the study and those who completed it, in terms of baseline data?
- The discussion section needs to be extensively revised. Very little comparison is made between the results of the present study and other published studies. There is a vast description of the mechanisms involved between zinc and the markers assessed, but the authors need to convince the reader of the importance of this study for clinical practice and the scientific literature.
- There is no "Conclusion" section in the study; as in the abstract, the objective of the study is not clearly addressed.
Reviewer 2 Report
Comments and Suggestions for Authors
1. In introduction, begin with a statement about the global impact or importance of zinc.
- Transition between concepts need to be more fluid in introduction. This will help to guide the reader through the importance of zinc, its deficiencies, and the interventions studied.
- Introduce the global and regional context of zinc deficiency earlier in the introduction to set the stage for the study’s relevance.
- Discuss the challenges in zinc measurement and why alternative biomarkers are needed in a clear, concise way in the introduction.
- End the introduction with a clear, concise statement of the study’s purpose, hypothesis, and expected impact.
- The description of the study design and setting could be expanded to provide more context. For example, it would be helpful to briefly mention why a cluster-randomized controlled trial was chosen and what specific hypotheses or outcomes were anticipated from this design. Additionally, describing why wheat biofortification is particularly suitable for this population would strengthen the rationale.
- Although ethical approval is mentioned, there could be more information about participant consent, assent (given the age of adolescent participants), and any measures taken to protect participant confidentiality and safety. This is important to ensure ethical transparency and adherence to standards for research involving minors.
- The criteria for inclusion and exclusion could be elaborated. For instance, did the study exclude individuals with certain health conditions or dietary restrictions?
- The sample selection process could benefit from additional detail on how the randomization was stratified by clusters and any steps taken to prevent potential biases.
- It would also be beneficial to explain why a subsample of participants was selected for oxylipin measures, as this could impact the generalizability of these findings.
- The Data and Sample Collection, Storage, and Transportation section of materials and method part should clarify if there were any pre-collection requirements for participants, such as dietary or medication restrictions, which could influence plasma biomarkers.
- In the Data and Sample Collection, Storage, and Transportation section more detail could be provided on how the plasma samples were monitored during transportation to UC Davis to ensure sample integrity.
- In the Data and Sample Collection, Storage, and Transportation section adding information on sample handling, such as how long samples were stored at different temperatures, would make this section more robust.
- In the Laboratory Methods section on lipid analysis, include more information about the validation and calibration of the instruments used for FA and oxylipin measurements.
- In the Laboratory Methods section, mentioning any quality control or repeat measurements performed to ensure accuracy would enhance the credibility of the results.
- In the Laboratory Methods section, the FA and oxylipin analysis description is somewhat terse; it might be helpful to explain the rationale for focusing on specific fatty acids and oxylipins and how these relate to zinc deficiency.
- The statistical analysis section could be strengthened by explaining why certain transformations (log or Johnson-transformation) were chosen and how outliers were handled.
- There is no mention of power analysis in statistical analysis to justify the sample size, which is important in clinical studies to determine if the study is adequately powered to detect significant differences.
- The criteria for assessing statistical significance are not fully explained. For example, adjustments for multiple comparisons might be necessary, given the number of biomarkers analyzed.
- The meaning of a “significant group × time interaction” in statistical analysis could be clarified to ensure that readers understand how intervention effects were determined.
- Some points are repeated in different parts of the discussion. For example, the importance of zinc biofortification and the relationship between zinc deficiency and health are mentioned multiple times. Streamline these points into one clear, concise statement at the beginning.
- The discussion could benefit from a clearer structure. Organizing it into sub-sections like "Key Findings," "Mechanisms," "Comparison with Other Studies," "Limitations," and "Implications" could make it easier for readers to follow the argument.
- The main findings, such as the increase in DHA and decrease in pro-inflammatory oxylipins, could be highlighted more prominently at the start of the discussion. This will help readers immediately grasp the significance of the results.
- In discussion, while background information on zinc and oxylipins is valuable, it could be condensed to avoid overshadowing the actual findings of the study. Focusing more on how the current study's results advance our understanding would strengthen the discussion.
- The mechanisms through which zinc biofortification could affect FA and oxylipin levels, particularly in adolescents, are not well-explained in the discussion. Expanding on how zinc might influence FA metabolism and oxylipin pathways could provide deeper insights into the biological relevance of the findings.
- There are some comparisons with previous studies (e.g., Suh et al., 2022) in discussion, but these comparisons could be more direct and detailed. Explaining why differences or similarities occur, such as differences in dosage or population characteristics, would add depth to the analysis.
- Although some limitations are mentioned in discussion, they are not given sufficient emphasis or explored in depth. For example, the lack of data on seasonal dietary variations is noted, but other limitations, such as the study's reliance on plasma markers alone or the relatively modest increase in zinc intake, could be elaborated upon.
- In discussion at some places, the language is without sufficient evidence (e.g., statements about cardiovascular or reproductive health impacts). It would be more scientifically rigorous to limit speculation to areas where there is supportive data or to clearly indicate when hypotheses are being proposed for future research.
- The potential public health impact of zinc biofortification could be explored in greater detail in the discussion. Discussing how these findings could inform policy or public health strategies, particularly in zinc-deficient regions, would enhance the relevance of the study.
- Some sentences are ambiguous in discussion, making them difficult to interpret. For example, the statement about the seasonality of plasma lipids could be simplified to clarify its relevance to the study results.
The quality of English is generally good and understandable, especially for an academic audience familiar with scientific writing. However, readability could be improved by simplifying sentence structure, reducing repetitions, making transitions more smooth, and maintaining consistent terminology. With these adjustments, the clarity and flow of the writing would likely improve, making the paper more accessible and engaging for a broader scientific audience.
Round 2
Reviewer 1 Report
Comments and Suggestions for Authors
Dear authors, thank you for adressing all comments.
Author Response
Thank you for your comments and suggestions, which helped us to improve the quality of our manuscript!
Reviewer 2 Report
Comments and Suggestions for Authors
The Manuscript can now be accepted for publication
Author Response

(The authors gave the same response as above.)
